# A proposed cigarette emissions topography protocol reflecting smokers' natural environment use behavior

Edward C. Hensel ◉⊚*, Risa J. Robinson⊚

Department of Mechanical Engineering, Rochester Institute of Technology, Rochester, New York, United States of America

⊚ These authors contributed equally to this work.

* echeme@rit.edu

## Abstract

### Background

The FTC, in 2008, rescinded its 1966 guidance regarding use of the Cambridge Filter Method, noting the yields from the method are relatively poor indicators of tar, nicotine, and carbon monoxide exposure. This article proposes a set of puffing conditions for cigarette emissions testing, with the goal of developing product-specific emissions characterizations which can subsequently be used to realistically model the yield of particulate matter and constituents to the mouth of a smoker, while accounting for the actual puffing behavior of the smoker.

### Methods

Synthesis of data was conducted on data collected from a prior one-week observation of 26 adult cigarette smokers, using their usual brand cigarette in each smokers' natural environment including the puff flow rate, duration, volume and time of day of each puff taken were recorded with a cigarette topography monitor. Data was analyzed to determine the empirical joint probability function and cumulative distribution function of mean puff flow rate and puff duration. The joint CDF was used to define an emissions topography protocol using concepts common to computational grid generation.

### Results

Analysis of 8,250 cigarette puffs indicated the middle 95% of mean puff flow rates varied between 15 and 121 [mL/s] while the middle 95% of puff duration varied from 0.55 to 3.42 [s].

### Conclusions

Thirteen conditions of varying mean puff flow rate and puff duration are proposed for a comprehensive cigarette emissions topography protocol. The proposed protocol addresses

**Data Availability Statement:** The data is provided as supplementary data accompanying the manuscript.

**Funding:** Funding: Research reported in this publication was supported by the United States'

National Institute of Environmental Health Sciences (NIEHS) of the National Institutes of Health and FDA Center for Tobacco Products (CTP) under award number R21ES029984. The content is solely the responsibility of the authors and does not necessarily represent the official views of the NIH or the Food and Drug Administration. This work was also funded by the United States' National Institute on Drug Abuse/National Institutes of Health (NIDA/NIH) and the Food and Drug Administration/Center for Tobacco Products (FDA/CTP) under grant number 1R01-DA042470-01. The content is solely the responsibility of the authors and does not necessarily represent the official views of the NIH or the FDA.

**Competing interests:** The authors have declared that no competing interests exist.

inadequacies associated with common machine-puffing profiles used for generating cigarette emissions.

## Introduction

The Federal Trade Commission (FTC) published in 1967 and revised in 1980 the "Cambridge Filter Method" as the standard for testing of cigarettes [1, 2], which was adopted by the International Standards Organization (ISO) as ISO 3308, and is widely known as the FTC/ISO protocol [3]. The FTC/ISO protocol specifies a machine which uses a motor driven, hydraulically-operated syringe device to produce a half-sine or bell-shaped puff with a nominal puff volume of 35 [mL], puff duration of 2 [s] and a puff taken every 60 [s] with no ventilation blocking. This protocol provided a standard by which emissions from different products and manufacturers could be compared under similar machine puffing conditions. However, the standard obfuscated a true comparison of emission under actual use conditions.

In 2001, the National Cancer Institute concluded that emissions generated from the FTC/ISO puffing protocol did not represent actual yield because the puffing protocol did not represent the way people smoke [4]. The FTC issued a notice in December of 2008, that rescinded its '1966 guidance that it generally is not a violation of the FTC Act to make factual statements of the tar and nicotine yields of cigarettes when statements of such yields are supported by testing conducted pursuant to the Cambridge Filter Method" [5]. Since 2008, manufacturers have not been permitted to make claims that one product is less harmful than another based on the FTC/ISO protocol. However, the FTC/ISO protocol is still cited in the 2012 FDA draft guidance for reporting HPHCs to represent the "non-intense" smoking regimen (77FR20030) [6].

Most U.S. States had adopted the FTC/ISO standard, with the notable exceptions of Massachusetts and Texas. The Massachusetts Department of Public Health (MDPH) modified the FTC/ISO test conditions to increase puff volume from 35 [mL] to 45 [mL], change the puff frequency from one puff every 60 [s] to one puff every 30 [s], retain the puff duration at 2 [s] and specify that 50% of the ventilation holes must be blocked [7]. The MDPH test method remained in effect as of February 2, 2018 according to the MDPH government website. Health Canada implemented their own test methods which required a puff volume of 55 [mL], duration of 2 [s] and one puff every 30 [s] with all ventilation holes must be blocked. The Health Canada protocol is cited in the 2012 FDA draft guidance for reporting HPHCs to represent "intense" smoking regimes (77FR20030) [6].

In 2011, the FDA called for consideration of realistic topography in their guidance for substantial equivalence [8] of tobacco products, but data continue to be lacking to inform standards based on realistic topography across the range of real use behavior. Similarly, the Canadian government enacted extensive revisions to their tobacco regulatory policy [9] specifically related to product reporting in 2019. The 2020 article by Robinson *et al.* [10] clearly demonstrated that none of the currently used puffing protocols (FTC/ISO, HC, MDPH) accurately reflected the topography of cigarette users in their natural environment [10].

Further, none of the previously proposed cigarette emissions testing methods account for the dependence of emissions, including Total Particulate Matter (TPM), nicotine, or other Hazardous and Potentially Hazardous Constituents (HPHCs) on the puffing topography (flow rate, duration and period) or how the cigarette is actually used. To date, there is no widely recognized standard for a machine puffing protocol which reflects the actual cigarette product use behavior of smokers.

An empirical model of tobacco product emissions was used to investigate the joint impact of user topography [11] and tobacco product characteristics on electronic cigarette emissions [12, 13] and water pipe tobacco products [14–16]. The empirical model is introduced here for combustible cigarettes. The Aerosol Constituent (AC) mass concentration, $C_{AC}$, was expressed as the product of the Total Particulate Mass (TPM) concentration, $C_{TPM}$, and the mass ratio, $f_{AC}$, of each constituent relative to the TPM. The empirical model was extended and validated [13] to predict the cumulative mass yield of each aerosol constituent, $Y_{AC}$, as the summation of yield per puff delivered to the mouth of an electronic cigarette user as the product of the flow rate dependent TPM mass concentration, constituent mass ratio, and puff volume:

$$Y_{AC} = \sum_{n=1}^{N} C_{TPM}(q, d) \times f_{AC}(q, d) \times v \qquad (1)$$

Eq (1) provides a quantitative basis for comparing the yield of various aerosol constituents as a function of users' topography characteristics, puff flow rate q, puff duration d, and puff volume v. The mass concentration, $C_{TPM}$, and the mass ratio, $f_{AC}$, may be empirically quantified for each tobacco product of interest.

This work proposes an emissions topography protocol (ETP) for combustible cigarettes to adequately assess emissions from cigarettes spanning the range of cigarette users' topography behavior in their natural environment. The empirical model (1) lays the foundation for quantitative comparison of emissions generated from different brands of combustible cigarettes, provides a framework for consistent emissions reporting across research labs, and ability to quantify relative harm potential between products.

This work addresses three fundamental gaps in the literature. First, prior work demonstrated that standard puff profiles, individually and collectively, failed to represent the range of user puffing behavior observed in the natural environment [10]. Second, it has been reported that benchtop emissions testing conducted using those standard puff profiles do not provide realistic estimates of emissions from cigarettes. Third, there is no established mathematical basis for comparing the relative emissions, and hence harm potential, between classes of tobacco products. This paper proposes a cigarette emissions topography protocol spanning the range of observed smoking behavior to be used for emissions studies as a replacement for previously used machine cigarette puffing protocols. The paper directly addresses the first two gaps and lays a foundation for addressing comparative emissions testing between cigarette products, and eventually, for comparing relative harm potential between cigarettes and other tobacco and inhaled nicotine products.

## Data set and analysis methods

Establishing an ETP for combustible cigarette emissions testing requires (1) a comprehensive data set of human subject cigarette smokers' puff topography in their natural environment, (2) conducting synthesis analysis of data set, and (3) formulating the proposed cigarette ETP.

### Data collection method

The natural environment observation study which yielded the data set for this analysis was previously described by [10]. The study protocol was reviewed and approved by the Rochester Institute of Technology (RIT) Institutional Review Board (IRB). All participants enrolled in the study completed a signed informed consent document. Briefly, prospective participants were screened using an on-line tool and subsequently invited to an intake appointment. Upon confirming eligibility and obtaining consent, each participant was loaned a battery-operated second-generation cigarette wPUM™ topography monitor to use in their natural environment, in

conjunction with their own-choice cigarettes, for a one-week observation period. Participants completed product-use questionnaires at intake and exit and were asked to keep a daily log of cigarette brand choice and non-compliance to note times they may have smoked without using the monitor. The participant was asked to turn on the second generation monitor prior to smoking each cigarette and turn off the monitor after finishing each cigarette. The previously reported data set included puff duration, volume, inter-puff interval and mean puff flow rate of N = 8,170 discrete puffs from 27 adult human subjects (over 18 years of age) consisting of 23 males and 3 females, with a median age (standard deviation) of 27 (7) years. The previous analysis reported the inter-puff interval between each puff within each monitor power-on window and reported the final interval within each observation window the average of the intervals during each monitor power-on window. Thirteen own-choice cigarette brands were smoked by participants during the observation period. Responses to the PhenX Tobacco Use Survey [17] indicated participants in the study cohort included cigarette only users, dual users of cigarette and water pipe, dual users of cigarette and electronic cigarettes, former users of water pipe, and former users of electronic cigarettes. Responses to the cigarette nicotine dependence questionnaire (NDQ) [17, 18] ranged from 1 to 16 with a mean (standard deviation) of 7.9 (4.2), suggesting the participant cohort spanned the range from no dependence to high dependence.

The data set is considered an inclusive sample of smoker's topography behavior in their natural environment, without recruitment limitations on age, sex, race, nicotine dependence, dual-use status or nicotine dependence. Previously proposed emissions topography conditions by FTC/ISO [1–3], the Massachusetts Department of Public Health (MDPH) [7], and Health Canada (HC) [6] do not specify the demographics, brand, or product use-frequency to which the standards apply [10], and do not accurately represent the range of mean of smoking behavior observed in the natural environment [10].

## Secondary data analysis method

The secondary data analysis method employed herein began from the raw monitor data files collected during each monitor power-on window for each participant. The secondary analysis resulted in the identification of n = 8,250 discrete puffs, while n = 8,170 discrete puffs were reported previously. The difference in puff count ($\Delta n = 80$) between the prior and current analysis was related to one participant who declared dual use of cigarettes and marijuana (using the monitor) on one observation day. All puffs were included in this secondary analysis, while some data was redacted previously. The difference in results was not significant. All time history data from each participant was sorted in order of increasing date/time stamp, including intervals when the monitor remained in power-on status between cigarettes, reflecting the likely event the participant forgot to turn the second generation monitor off. A minimum puff duration of 0.5 [sec] was required for an event to be recorded as a puff, and any flow rate excursions within a 0.5 [sec] window were consolidated into one puff. In this way, an initial inhale, brief pause of perhaps 0.1 [sec] and a subsequent inhale were counted as a single puff. In this secondary analysis, the "post puff interval" was computed as the time following the end of one discrete puff and preceding the beginning of the next discrete puff, spanning the entire one-week observation period of each participant. The final puff of each participant, at the end of their observation week, was assigned a non-numeric value and excluded from post-puff interval analysis. Thus, the over-night sleep period would be reported as a puff with a particularly long post-puff interval, and moderately long post-puff intervals were reflective of gaps between smoking sessions. Any puff separated from its preceding neighbor by more than 5 minutes was declared to be part of a new smoking session. Conversely, any puff within 5 minutes of neighboring puffs was associated with single smoking session. The previous analysis

correctly computed the inter-puff interval within each power-on session of the monitor, but incorrectly estimated the inter-puff interval between monitor power-on sessions, occasionally resulting in a small negative interval. The current secondary analysis defined a post-puff interval and was found to be robust and resulted in no spurious estimates. Both analyses produce identical results for puff duration, flow rate and volume. Data for every discrete puff measured and recorded for each study participant represented in the data set was concatenated into a "Discrete Puff Table (DPT)" (A1 Table in S1 File) for a secondary analysis consisting of 8,250 discrete puffs. Each record in the data set included a unique record identifier, a subject/participant number, puff duration (s), mean puff flow rate (mL/s), puff volume (mL), post-puff interval (s), time of day (seconds past midnight), and day of the week. The methods used to produce the data entered into the DPT closely follow the algorithm described previously in [10, 19] as modified herein for the secondary analysis.

All DPT data were plotted on a scatter plot of puff duration versus puff flow rate. Analysis of the DPT data was conducted to compute the empirical marginal cumulative distribution function (mCDF) as a function of puff flow rate, puff duration and puff volume denoted as mCDF(q), mCDF(d) and mCDF(v), respectively. The marginal CDFs were computed without segregating data by participant, cigarette brand, age, dependence, or any other demographic attribute. The mean puff flow rate, q, depends on the puff volume and duration by the relation $q = v / d$. While the mCDF for flow rate, duration and volume are not independent, it is sometimes useful to present all three sets of information on graphical displays of data. The marginal CDFs for each parameter were analyzed to determine the middle 95% of the data for flow rate, duration and volume as ranging between the 2.5th and 97.5th percentiles of each mCDF respectively. We also computed the 50th, 25th and 75th percentile of each mCDF to characterize the median and inter-quartile ranges of the topography parameters. A bounding box of the prevalent topography conditions was super-imposed on the scatter plot of all puff data by including those puffs which were above the 2.5th percentile of puff flow rate, duration and volume and also below the 97.5th percentile thereof. While the precise fraction of puffs, f, residing within the bounding box must be computed numerically, a first approximation is $f \approx 0.95^3 = 0.857 \approx$ 86%; suggests we should observe approximately 7,073 puffs inside the bounding box.

In addition to the scatter plot, mCDFs, and bounding box, an analysis was conducted on the sampling distribution of the mean (single observation per participant) for each of the topography parameters of puff flow rate, duration, volume, interval and puff count to determine the cohort mean values, standard deviations, and 95% confidence intervals.

## Procedure for formulating the emissions topography protocol

A topography envelope was derived to encompass the puff topographies of approximately central 86% of all cigarette puffs observed among the study cohort using the bounding box as described above. Next, it was desired to establish a sequence of discrete puff topography conditions which comprised the ETP for emissions testing of combustible cigarettes. The ETP may specify an arbitrary number of flow conditions.

The ETP is based on the concept of conformal grid mapping commonly employed in finite element analysis and other computer simulations of physical phenomena. To illustrate the underlying idea, consider a dimensionless topography domain, which spans the entire range of puff flow rates and durations exhibited by a cohort of tobacco product users in their natural environment. In this case, we can think of the space as bounded by the marginal CDFs of flow rate and duration which naturally range from 0 to 1, as illustrated in Fig 1. The domain is spanned by thirteen flow conditions, or nodes, as denoted by the circles. Each flow condition represents a particular flow rate and duration which is representative of the population data

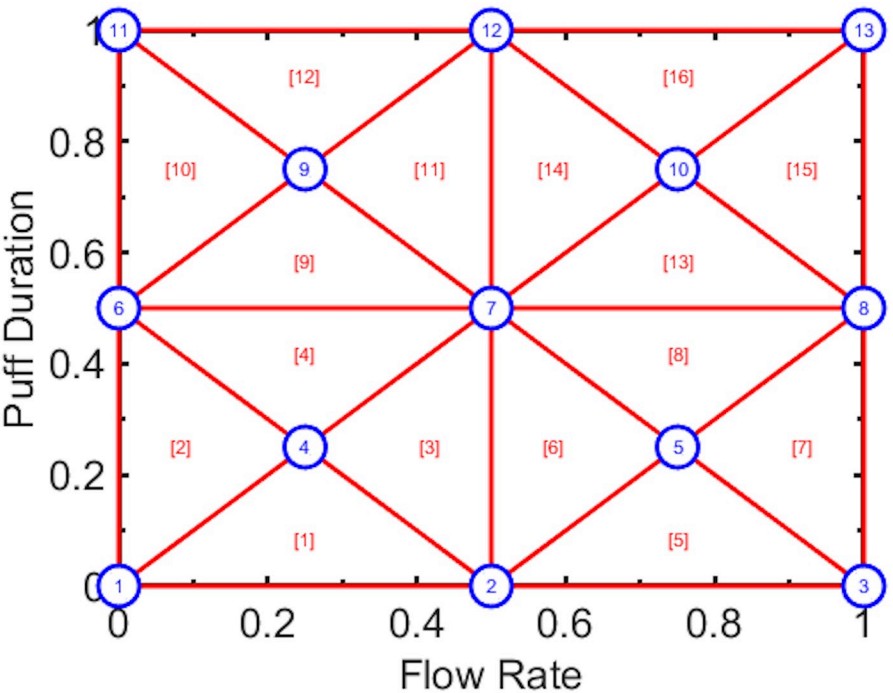

**Fig 1. Prototypical emission topography protocol composed of 13 flow conditions (pattern 1-3-5-3-1).** The horizontal and vertical axes represent the marginal cumulative distribution functions of natural environment observed puff flow rate and duration, respectively.

set. For example, node 7 represents the 50th percentile (the median) flow rate and duration, while node 13 represents the 100th percentile (maximum flow rate and duration) of exhibited puffs.

For the case shown in Fig 1, each triangular area encompasses approximately 6.25% of the discrete puff population. Emissions testing trials may be conducted using a machine puffing systems, in a manner similar to the testing procedures, sample collection and analytical chemistry procedures described in FTC [2] / ISO [3], MDPH [7], HC [20] and elsewhere. After emissions testing is conducted, the TPM mass concentration, $C_{TPM}$, and mass ratio of all aerosol constituents of concern, $f_{AC}$, may be empirically estimated. The emissions at any arbitrary flow condition may be estimated using linear regression as described in [13] or using bilinear interpolation between the arbitrary flow condition's nearest neighbors. For example, if it is necessary to estimate the aldehyde yield at the 25th percentile of flow rate and the 90th percentile of duration, the weighted average of $C_{TPM}$ and $f_{aldehyde}$ at nodes 9, 11, and 12 can be used to interpolate the mass concentration and mass ratio at the centroid of the triangle, and subsequently apply Eq (1) to compute the yield.

The conditions shown in Fig 1 include many trivial conditions: nodes 1, 6, and 11 have no flow while nodes 1, 2, and 3 have no duration—and hence no emissions. So, instead of extending the flow conditions to span the entire range of CDF from 0 to 1, any acceptable subdomain can be specified. For example, nodes 1, 6, 11 may be aligned along the 2.5th flow rate percentile and nodes 3, 8, 13 along the 97.5th flow rate percentile. Similarly, nodes 1, 2, 3 may be aligned along the 2.5th puff duration percentile and nodes 11, 12, 13 along the 97.5th duration percentile. The remaining nodes 4, 5, 7, 9, 10 can be uniformly distributed across the domain. We refer to this thirteen condition pattern as 1-3-5-3-1, as illustrated in Fig 1, to indicate there are

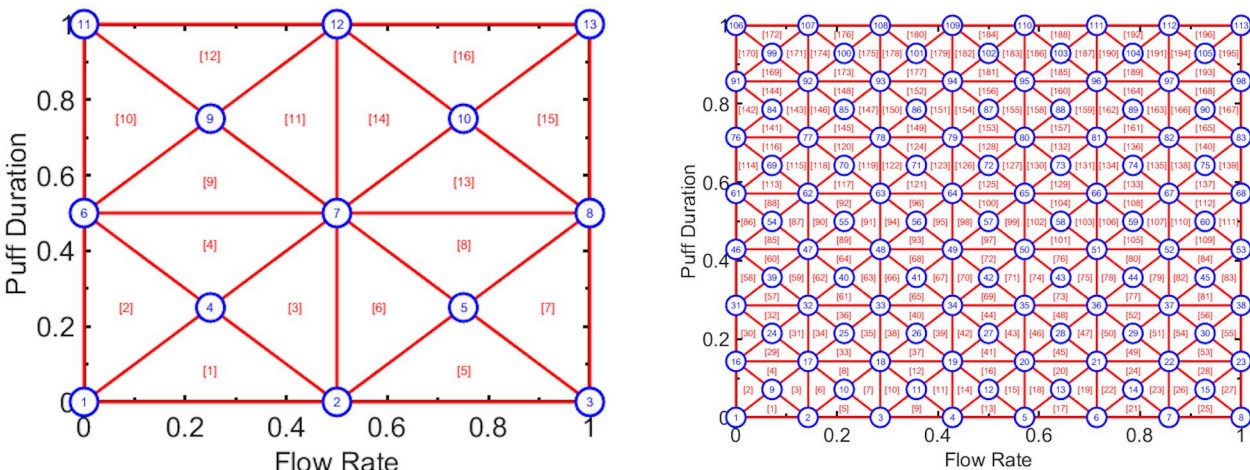

**Fig 2. High fidelity emission topography protocols composed of 61 flow conditions (left panel, pattern 1-3-5-7-9-11-9-7-5-3-1) and 113 flow conditions (right panel, pattern 1-3-5-7-9-11-13-15-13-11-9-7-5-3-1).** The horizontal and vertical axes represent the marginal cumulative distribution functions of natural environment observed puff flow rate and duration, respectively.

5 flow conditions along the main diagonal (nodes 1,4,7,10,13). Now consider the area of each triangle bounded by any combination of three nearest neighbor nodes. The triangles are denoted by the red lines and uniquely named by the number in square brackets. For example, triangle [9] is composed of nodes 6, 7, 9 while triangle [16] is composed of nodes 10, 13, 12. The entire area enclosed by the triangles represents 100% of the range of all puffs exhibited in a data set. If the left, bottom, right and top borders are aligned to enclose the central 95% of puff flow rates and durations, the entire area would enclose 90% of all puffs in a data set, and each triangle would represent approximately 5.6% of the puff population.

The number of flow conditions employed in the ETP may be increased to obtain higher fidelity representations of the emissions surface, with the corresponding time and expense of additional emissions trials. This is illustrated in Fig 2, which shows a high fidelity ETP with 61 flow conditions and 11 nodes along the main diagonal, and a very high fidelity ETP with 113 flow conditions and 15 nodes along the main diagonal. Each triangular area of the left ETP represents 1% of the population of discrete puffs, while each triangle on the right represents approximately 0.5% of the puffs.

In practice of formulating an ETP, it is also necessary to enforce a constraint on the upper limit of puff volume, usually at the 97.5[th] percentile of puff volume, or at a physical limitation of maximum achievable volume. As is well known, the puff volume is nominally the product of puff duration and puff flow rate. However, the observed maximum puff duration times the observed maximum puff flow rate may give rise to an unrealistically large puff volume. This maximum puff volume constraint may be a function of tobacco product category. For example, the maximum ETP puff volume may be limited by the maximum inspiratory volume for water pipe users and direct to lung electronic cigarette users. Conversely, the ETP puff volume may be limited by the maximum oral cavity volume or the length of a cigarette for cigarette smokers.

## Results

Fig 3 provides a scatter plot of N = 8,250 cigarette puffs taken by the 26 smokers during the course of one-week observation periods in their natural environment. The analysis presented in Fig 3 is based on the underlying data previously summarized in [10]. Each green circle in

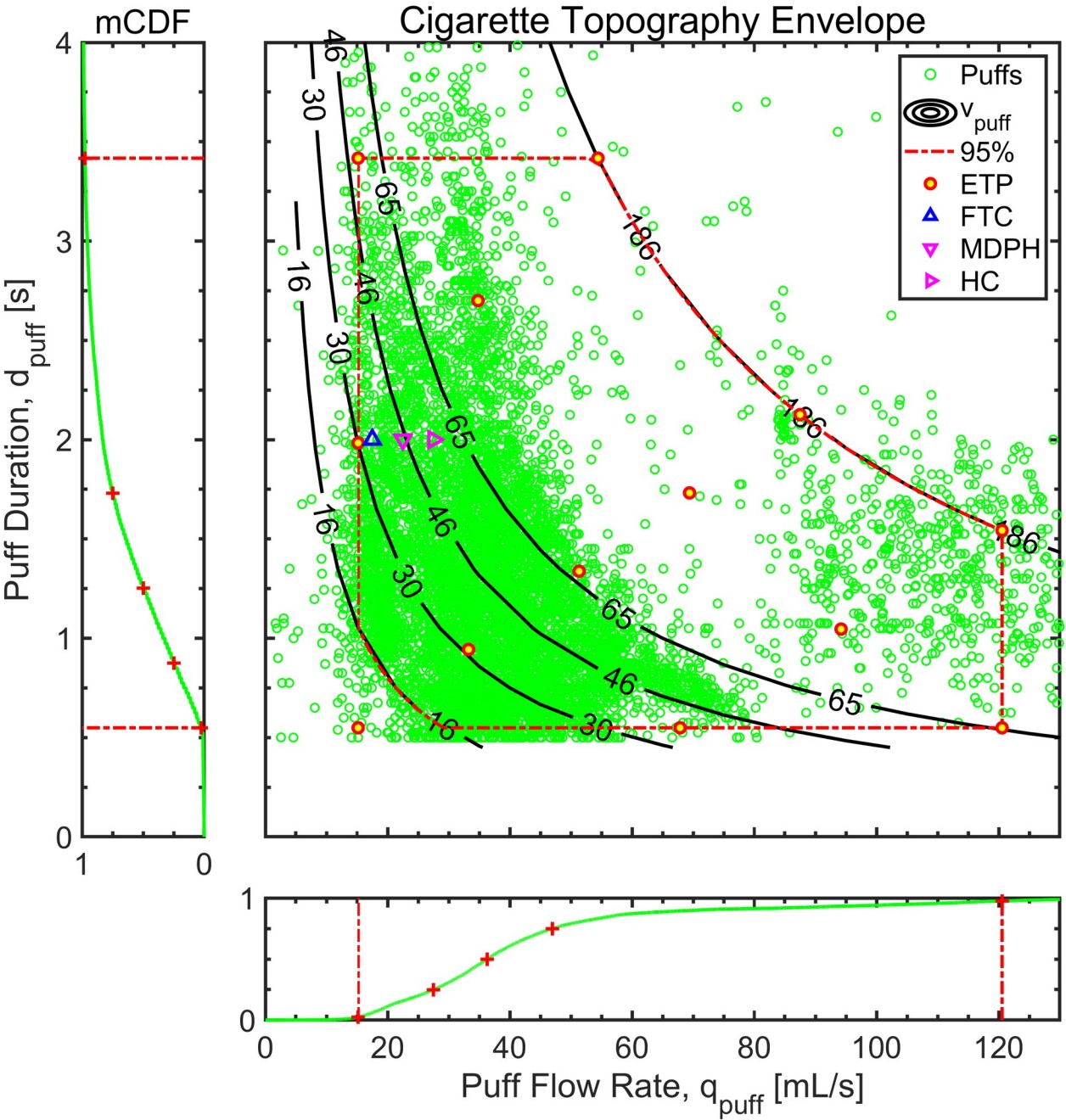

**Fig 3. Puff topography distribution (N = 8,250 puffs) of (N = 26 smokers) of cigarette smokers in their natural environment during a week-long observation period.** The '+' represent 2.5th, 25th, 50th, 75th, and 97.5th percentiles in the marginal cumulative distribution of puff duration and flow rate. Lines of constant volume are shown for the 2.5th, 25th, 50th, 75th, and 97.5th in the mCDF of puff volume. The red circles illustrate the recommended emissions topography protocol (ETP) representing smokers in the natural environment. Participants ranged from occasional smokers (less than daily) to an upper limit of ½ pack per day.

the main panel represents the puff duration and mean puff flow rate of one discrete puff taken by one study participant. The puff volumes are shown by lines of constant volume superimposed on the plot. The raw data for every puff is available as supplemental data file A. The sampling distribution of the mean (single observation per participant) exhibited mean (standard

**Table 1. Proposed emissions topography protocol reflecting smoker's use behavior in their natural environment for cigarette emissions characterization testing.**

| Condition | Puff Flow Rate [mL/s] | Puff Duration [s] | Puff Interval [s] | Minimum Number of Trials [-] |
|:---:|:---:|:---:|:---:|:---:|
| 1 | 15 | 0.5 | 21 | 6 |
| 2 | 121 | 0.5 | 21 | 6 |
| 3 | 15 | 3.4 | 21 | 6 |
| 4 | 121 | 1.5 | 21 | 6 |
| 5 | 54 | 3.4 | 21 | 6 |
| 6 | 87 | 2.1 | 21 | 6 |
| 7 | 33 | 0.9 | 21 | 6 |
| 8 | 51 | 1.3 | 21 | 6 |
| 9 | 69 | 1.7 | 21 | 6 |
| 10 | 68 | 0.5 | 21 | 6 |
| 11 | 15 | 2.0 | 21 | 6 |
| 12 | 94 | 1.0 | 21 | 6 |
| 13 | 35 | 2.7 | 21 | 6 |

deviations) of 1.5 (0.59) [s] puff duration, 40.4 (9.8) [mL/s] puff flow rate, 55.3 (19.7) [mL] puff volume, and 35 (132) [s] puff interval. The topography conditions reflecting the FTC [1, 2], MDPH [7] and HC [20] puff protocols are shown in Fig 3 as 2 [s] duration puffs with effective mean puff flow rates of 17.5, 22.5, and 27.5 [mL/s], respectively. It is clear that these standards do not span the range of actual smoking behavior, as previously reported [10]. The upper region of the ETP was constrained to a maximum puff volume of 186 [mL], representing the 95th percentile of observed puff volumes, and also reflecting a reasonable physiological limit on oral cavity volume for smokers who use a mouth-to-lung puff maneuver. The upper right node 13 of Fig 2 was positioned along the middle of the maximum puff volume arc, while nodes 12 and 8 were positioned at the intersection of the maximum puff volume constraint and the 95th percentile of puff duration and flow rate, respectively. Nodes 6 and 2 were positioned at the mid-point of the puff duration and puff flow rate range, respectively.

The empirical mCDF of puff duration is shown along the left border panel of Fig 3 and the mCDF of puff flow rate is shown along the bottom border panel. The 2.5th, 25th, 50th, 75th, and 97.5th percentiles for puff flow rate were 15, 28, 36, 47, and 121 [mL/s], for puff duration were 0.6, 0.9, 1.3, 1.7, and 3.4 [s], and for puff volume were 16, 30, 46, 65, 186 [L].

The combination of puffing conditions, labeled as ETP in Fig 3 **and listed in** Table 1, are the recommended topography conditions for the ETP representative of the NE cigarette smoking. Completion of this protocol (13 conditions and 6 repeated trials) will require 78 cigarettes (about 4 packs of 20) and will result in a surface map describing the range of emissions as a function of topographies observed in the NE.

The median (50th percentile) of inter-puff interval was observed to be 16 [s], while the mean was computed as 35 [s] and the mode was 11 [s]. The proposed ETP recommends an interval between puffs of 21 [s] (puff end to puff start) such that the effective puff period (puff start to puff start) ranges from 21.6 [s] to 24.4 [s]. The proposed puff interval is slightly shorter than the 30 [s] puff period specified by MDPH [7] and substantially shorter than the puff period previously established in the FTC method [1, 2].

## Discussion

The secondary data analysis reported here resulted in n = 8,250 puffs with durations ranging from 0.5 to 10.0 [sec] and puff volumes ranging from 1.3 to 600 [mL]. 97.5% of all puffs had a volume less than or equal to 186 [mL] and duration of less than 3.5 [sec]. While the long

durations and puff volumes may appear physiologically unrealistic for a mouth-to-lung puffing maneuver of cigarette smoking, this range of behaviors was observed with the monitor. Possible conjectures for such observations could include users inhale through the monitor before starting or after finishing a cigarette, or ineffectively seating the cigarette in the monitor, and permitting air flow around the perimeter of the filter. These observations reflected a very small fraction of the observed puffs; all puffs were reported in the data set in an effort to permit the research community to independently assess their importance, relevance, and possible source.

The working hypothesis that natural environment observations of cigarette smoker's topography behavior can be used to inform an emissions topography protocol for cigarette emissions testing has been demonstrated.

The method for defining the ETP is flexible and may be applied to create emissions testing conditions representative of the naturalistic behavior of any tobacco product. The method permits consideration of puff duration, flow rate and volume constraints consistent with observed user behavior, physiological constraints, and product operating envelopes. The ETP provides a robust testing envelope which by design reflects the range of observed user behavior. Developing an ETP using natural environment observation data for users of various tobacco products ensures both that emissions testing for different products is sufficient, and permits direct one-to-one comparisons between the behavior patterns of various tobacco product users.

The proposed behavior-based ETP for emissions testing overcomes the primary shortcomings of the deprecated FTC test method [1, 2] which was shown to be an insufficient reflection of product usage and not an accurate predictor of particulate matter or nicotine yield under realistic smoking conditions [5].

The proposed ETP may permit a comprehensive characterization of specific cigarette designs to better estimate the smoker-specific yields of particulate matter and nicotine [13] which may be anticipated and reflective of each smokers' natural environment use behavior. Should it be desired to achieve higher fidelity in emissions testing, the number of sample points along each marginal CDF may be increased.

The number of topography test conditions can be increased throughout the domain or can be focused on a subset of the domain known to represent the use behavior for a particular cigarette or clinical trial participant group. Likewise, as more collective knowledge about naturalistic cigarette smoking topography becomes available, the ETP sample points may be refined to reflect the emerging knowledge base. This general approach to experiment design may be extended to a variety of inhaled tobacco [14, 15] and other products using available topography data.

The method for determining the joint CDF of cigarette puff topography is broadly applicable to the analysis of any intensive observational study of puff topography in the natural environment. The specific data collected for this study was limited to a one-week natural environment observation period with a non-uniform distribution of males and females. The study cohort consisted predominantly of young adult smokers of less than ½ a pack of cigarettes per day. A further study with a larger cohort of a broader distribution of demographics and cigarette products may extend the broad applicability of the emission ETP. The ETP is illustrated for 13 flow conditions. The fidelity of the test method, as with any experimental protocol, may be improved by adding more test conditions and increasing the number of repeated trials.

Future directions for this work include:

- Use the Emissions Topography Protocol (ETP) to conduct emissions tests of one or more cigarettes representative of the participants' product choices.

- Apply the previously published framework [13] to estimate yield of TPM, nicotine and other aerosol constituents delivered to cigarette smokers [11] as a function of time.

- Validate the yield model for combustible cigarettes by this approach; (1) Combine the NE topography data and the emissions data to conduct "playback emissions studies" of the Marlboro smokers (2) Capture $Y_{TPM}$ and $Y_{nic}$ emissions from the playbacks (3) Compare the daily values of $Y_{TPM}$ and $Y_{nic}$ against the yield model.

## Conclusions

This paper addresses the inadequacies of current machine-driven cigarette emissions protocols to assess the full range of natural environment smoking behavior by proposing a design of experiments approach based on real usage data instead. Using a range of puff topographies which reflects naturalistic smoker behavior may lead to improved understanding of the joint relationships between user behavior and product characteristics on emissions yield, correlation with biomarkers of exposure, and ultimately on predicting health effects.

## Supporting information

**S1 File.** The following are available online at insert data link here, A1 Table: Discrete puff data for 8,250 cigarette puffs taken by 26 study participants during a week-long observation period in their natural environment. A2 Table: Participant demographics and descriptive statistics for 26 study participants during a week-long observation period in their natural environment. (XLSX)

## Acknowledgments

The authors wish to acknowledge the support and contributions of our colleagues in the Respiratory Technologies Lab: Dr. Nathan C. Eddingsaas, A. Gary DiFrancesco, Shehan Jayasekera, Bryan T. Meyers, A. A. Olayan, Qutaiba M. Saleh, S. Emma Sarles and Mahagani Thomas, and at the University of Rochester: Dr. Irfan Rahman.

## Author Contributions

**Conceptualization:** Edward C. Hensel, Risa J. Robinson.

**Formal analysis:** Edward C. Hensel, Risa J. Robinson.

**Funding acquisition:** Risa J. Robinson.

**Methodology:** Edward C. Hensel.

**Project administration:** Risa J. Robinson.

**Resources:** Risa J. Robinson.

**Software:** Edward C. Hensel.

**Validation:** Edward C. Hensel.

**Visualization:** Edward C. Hensel, Risa J. Robinson.

**Writing – original draft:** Edward C. Hensel.

**Writing – review & editing:** Risa J. Robinson.

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
