## [Decision Letter · Decision Letter 0]

24 Jan 2022

PONE-D-21-22198A proposed cigarette emissions topography protocol reflecting smokers’ natural environment use behavior via synthesis of previously reported dataPLOS ONE

Dear Dr. Hensel,

Thank you for submitting your manuscript to PLOS ONE. After careful consideration, we feel that it has merit but does not fully meet PLOS ONE’s publication criteria as it currently stands. Therefore, we invite you to submit a revised version of the manuscript that addresses the points raised during the review process.

Please, follow carefully the questions risen by referees, especially from the Reviewer #2.

We look forward to receiving your revised manuscript.

Kind regards,

Pasquale Avino, Ph.D.

Academic Editor

PLOS ONE

Journal Requirements:

2. Please amend either the title on the online submission form (via Edit Submission) or the title in the manuscript so that they are identical.

4. Please upload a copy of Figure 4, to which you refer in your text on page 12. If the figure is no longer to be included as part of the submission please remove all reference to it within the text.

5. We note that this manuscript is a systematic review or meta-analysis; our author guidelines therefore require that you use PRISMA guidance to help improve reporting quality of this type of study. Please upload copies of the completed PRISMA checklist as Supporting Information with a file name “PRISMA checklist”.

Reviewers' comments:

Reviewer #1

The paper aims at proposing a new protocol to test cigarettes in terms of emissions. This is an important aspect as it would allow a standardization of this kind of measurement.

I suggest to accept the paper after having included the following revisions:

- It is not clear how the parameters under investigation were measured. Actually, details on the experimental analysis are completely missing. This is maybe due to the fact that they were presented in a previous paper, nonetheless a quick recap of the experimental method is essential to support the reader to follow the paper.

- In my opinion, even if the topic was introduced through a very long introduction, the reader could not understand the importance of the proposed paper/topography. Did the results of characterization of the cigarette emission would change adopting previous or different protocols?

Reviewer #2

This manuscript analyzed cigarette smoking topography data from 26 participants and generate emission testing table to better assist regulating authorities. The authors well organized the manuscript in clear way, but there is a concern in the dataset. The authors should have cleaned the dataset before the analysis, but there are outliers in appendix dataset and the authors didn't mention this in the manuscript. Please see below comment. This manuscript could be published after address this issue.

L121: It would be helpful if the authors can provide more information about the topography data (e.g., wPUM topography monitor was used, etc.). In addition, did the authors curate or clean data? I am assuming the authors used the same criteria from the reference (Robinson at al., 2020), but the appendix dataset includes datapoints with negative puff intervals. I also observe puff duration ~10 sec and volumes over ~500ml. Do the authors think the values are reasonable? Is the appendix data raw file before cleaning? Descriptive statistics of the dataset, explanations, and better justifications should be added in this section.

Robinson, R. J., Sarles, S. E., Jayasekera, S., Al Olayan, A., Difrancesco, A. G., Eddingsaas, N. C., & Hensel, E. C. (2020). A Comparison between Cigarette Topography from a One-Week Natural Environment Study to FTC/ISO, Health Canada, and Massachusetts Department of Public Health Puff Profile Standards. International journal of environmental research and public health, 17(10), 3444.

Reviewer #2:** **Yeongkwon Son

---

## [Author Response · Author response to Decision Letter 0]

5 Feb 2022

Authors’ Response to Academic Editor - Pasquale Avino, Ph.D.

Editor Comment 1: A rebuttal letter that responds to each point raised by the academic editor and reviewer(s). You should upload this letter as a separate file labeled 'Response to Reviewers'.

Authors’ Response: This letter contains our response.

Editor Comment 2: A marked-up copy of your manuscript that highlights changes made to the original version. You should upload this as a separate file labeled 'Revised Manuscript with Track Changes'.

Authors’ Response: Confirmed.

Editor Comment 3: An unmarked version of your revised paper without tracked changes. You should upload this as a separate file labeled 'Manuscript'. 

Authors’ Response: Confirmed.

Editor Comment 4: Authors’ Response: No changes to financial disclosure are needed.

Editor Comment 5: If applicable, we recommend that you deposit your laboratory protocols in protocols.io to enhance the reproducibility of your results. Protocols.io assigns your protocol its own identifier (DOI) so that it can be cited independently in the future. For instructions see: https://journals.plos.org/plosone/s/submission-guidelines#loc-laboratory-protocols. Additionally, PLOS ONE offers an option for publishing peer-reviewed Lab Protocol articles, which describe protocols hosted on protocols.io. Read more information on sharing protocols at https://plos.org/protocols?utm_medium=editorial-email&utm_source=authorletters&utm_campaign=protocols.

Authors’ Response: While we have used protocols.io in the past, and hope to again in the future, we do not see the direct need to such a link related to the current manuscript. 

Editor Comment 6: Please, follow carefully the questions risen by referees, especially from the Reviewer #2.

Authors’ Response: Thank you. Our responses below address each comment in sequence. 

Editor Comment 7: Please submit your revised manuscript by Mar 10 2022 11:59PM. If you will need more time than this to complete your revisions, please reply to this message or contact the journal office at plosone@plos.org. Authors’ Response: Confirmed.

Authors’ Response to Journal Requirements

Journal Requirement 1: Please ensure that your manuscript meets PLOS ONE's style requirements, including those for file naming. The PLOS ONE style templates can be found at 

Authors’ Response: We have modified the section headings and author affiliations accordingly.

Journal Requirement 2: Please amend either the title on the online submission form (via Edit Submission) or the title in the manuscript so that they are identical.

Authors’ Response: We have used “Edit Submission” to reflect the title contained within the manuscript.

Journal Requirement 3: Please include your full ethics statement in the ‘Methods’ section of your manuscript file. In your statement, please include the full name of the IRB or ethics committee who approved or waived your study, as well as whether or not you obtained informed written or verbal consent. If consent was waived for your study, please include this information in your statement as well. 

Authors’ Response: We have edited the subsection “Data Collection Method” to include this statement.

Journal Requirement 4: Please upload a copy of Figure 4, to which you refer in your text on page 12. If the figure is no longer to be included as part of the submission please remove all reference to it within the text.

Authors’ Response: There is no Figure 4, these references were a typographical error held over from a prior submission, which had included a PRISMA analysis and guidance based on the original title. We have corrected all references to “Figure 4” which should have been references to “Figure 3”.

Journal Requirement 5: We note that this manuscript is a systematic review or meta-analysis; our author guidelines therefore require that you use PRISMA guidance to help improve reporting quality of this type of study. Please upload copies of the completed PRISMA checklist as Supporting Information with a file name “PRISMA checklist”. 

Authors’ Response: The PRISMA checklist previously submitted with Manuscript Version #2 has been uploaded as Supporting Information with a filename “PRISMA Checklist”. This was addressed during the original submission process and PLOS ONE staff editorial review. Our first version of the manuscript (#1), titled “A proposed cigarette emissions topography protocol reflecting smoker’s natural environment use behavior”, submitted on July 7, 2021 did not include a PRISMA analysis and figure. A copy of that figure is shown here for reference. After submitting version (#2), in August 2021, titled “A proposed cigarette emissions topography protocol reflecting smokers’ natural environment use behavior via meta-analysis of previously reported data”, the PLOS ONE staff agreed our analysis was a synthesis of a single data set, and did not actually require the PRISMA checklist. At that point, we removed the PRISMA text, and uploaded the revised the manuscript again (#3) in September 2021. The title of manuscript (#3) was reverted to “A proposed cigarette emissions topography protocol reflecting smoker’s natural environment use behavior” reflecting guidance of PLOS ONE staff, but we failed to amend the title using the “Edit Submission” tool. Manuscript (#3) was the version sent to peer reviewers, and we have now revised in accordance with our response to reviewers (Version #4). The PRISMA related text, presented in version #2, and removed in version #3 at the request of PLOS ONE staff, is provided below for consideration by the Academic Editor. We remain unclear about the need for the PRISMA checklist and the figure. We did not include it (#1), so we put it in (#2), then were asked to remove it (#3), which we did, and now are being asked to put it back in (#4). Unfortunately, the manuscript was not assigned to you as the Academic Editor until December 2021, and progress has proceeded swiftly since then. Thank you! We are perfectly happy to either include or exclude the PRISMA checklist and / or figure. We simply need clear guidance how to proceed. We edited the text in section “Data Synthesis” method to clearly state this manuscript is a “secondary analysis” and renamed the subsection as “Secondary Data Analysis Method.”

Version #2 manuscript (following PLOS staff guidance) contained this text: 

2.2. Meta-Analysis Method

Data for every discrete puff measured and recorded for each study participant represented in the data set was concatenated into a “Discrete Puff Table (DPT)” (Supplemental Data File A1) of 8,250 puff records as indicated by the PRISMA flow chart shown in Figure 1. … 

Version #3 manuscript (as reviewed) contained this text (following revised guidance):

2.2 Data Synthesis Method

Data for every discrete puff measured and recorded for each study participant represented in the data set was concatenated into a “Discrete Puff Table (DPT)” (Supplemental Data File A1). …

Version #4 manuscript (response to peer review) contains:

2.2 Data Synthesis Method

Data for every discrete puff measured and recorded for each study participant represented in the data set was concatenated into a “Discrete Puff Table (DPT)” (Supplemental Data File A1) for a secondary analysis. … 

Text History. Version #2 (containing PRISMA analysis) and Version #3 (PRISMA analysis and checklist removed as requested by PLOS Staff prior to review), and Version #4 following peer review. Figure 1. This figure was submitted as part of a PRISMA analysis for manuscript Version #2, which was subsequently removed for manuscript Version #3 at the guidance of PLOS ONE Staff, prior to peer review.

---

## [Decision Letter · Decision Letter 1]

17 Mar 2022

A proposed cigarette emissions topography protocol reflecting smokers’ natural environment use behavior

PONE-D-21-22198R1

Dear Dr. Hensel,

We’re pleased to inform you that your manuscript has been judged scientifically suitable for publication and will be formally accepted for publication once it meets all outstanding technical requirements.

Kind regards,

Pasquale Avino, Ph.D.

Academic Editor

PLOS ONE

Reviewers' comments:

Reviewer #1: The authors have now properly addressed the comments, thus the paper can be accepted for publication.

Reviewer #2: The authors well addressed all editor's and reviewer's comments. I recommend accept this manuscript.

---

## [Editor Report · Acceptance letter]

28 Mar 2022

PONE-D-21-22198R1 

A proposed cigarette emissions topography protocol reflecting smokers’ natural environment use behavior 

Dear Dr. Hensel:

I'm pleased to inform you that your manuscript has been deemed suitable for publication in PLOS ONE. Congratulations! Your manuscript is now with our production department. 

Kind regards, 

on behalf of

Professor Pasquale Avino 

Academic Editor

PLOS ONE